# Functional Characterization of the GNAT Family Histone Acetyltransferase Elp3 and GcnE in *Aspergillus fumigatus*

**DOI:** 10.3390/ijms24032179

**Published:** 2023-01-22

**Authors:** Young-Ho Choi, Sung-Hun Park, Sung-Su Kim, Min-Woo Lee, Jae-Hyuk Yu, Kwang-Soo Shin

**Affiliations:** 1Department of Microbiology, Graduate School, Daejeon University, Daejeon 34520, Republic of Korea; 2Department of Biomedical Laboratory Science, Daejeon University, Daejeon 34520, Republic of Korea; 3Soonchunhyang Institute of Medi-Bio Science, Soonchunhyang University, Cheonan 31151, Republic of Korea; 4Department of Bacteriology, University of Wisconsin-Madison, Madison, WI 53706, USA

**Keywords:** *Aspergillus fumigatus*, histone acetyltransferase, GNAT family, biofilm formation, stress responses, secondary metabolites, virulence

## Abstract

Post-translational modifications of chromatin structure by histone acetyltransferase (HATs) play a pivotal role in the regulation of gene expression and diverse biological processes. However, the function of GNAT family HATs, especially Elp3, in the opportunistic human pathogenic fungus *Aspergillus fumigatus* is largely unknown. To investigate the roles of the GNAT family HATs Elp3 and GcnE in the *A. fumigatus*, we have generated and characterized individual null Δ*elp3* and Δ*gcnE* mutants. The radial growth of fungal colonies was significantly decreased by the loss of *elp3* or *gcnE*, and the number of asexual spores (conidia) in the Δ*gcnE* mutant was significantly reduced. Moreover, the mRNA levels of the key asexual development regulators were also significantly low in the Δ*gcnE* mutant compared to wild type (WT). Whereas both the Δ*elp3* and Δ*gcnE* mutants were markedly impaired in the formation of adherent biofilms, the Δ*gcnE* mutant showed a complete loss of surface structure and of intercellular matrix. The Δ*gcnE* mutant responded differently to oxidative stressors and showed significant susceptibility to triazole antifungal agents. Furthermore, Elp3 and GcnE function oppositely in the production of secondary metabolites, and the Δ*gcnE* mutant showed attenuated virulence. In conclusion, Elp3 and GcnE are associated with diverse biological processes and can be potential targets for controlling the pathogenic fungus.

## 1. Introduction

Histone acetyltransferases (HATs) catalyze the acetylation of lysines within the N-terminal tails and globular domains of histone proteins by transferring an acetyl group using acetyl-CoA as a substrate. By this modification, histones form transcriptionally active euchromatin structure, enabling active gene expression [1,2,3,4]. These HATs are classified into two major groups based on their cellular locations and functions. Type A HATs are located in the nucleus and acetylate nucleosomal histones, whereas type B HATs are placed in cytoplasm and acetylate newly synthesized histones. Moreover, based on the conserved structural motifs, type A HATs can be further divided into five families, including GNAT (Gcn5-related N-acetyltransferases), MYST (MOZ, MOF, MORF, YBF2/SAS3, SAS2, TIP60, and HBO1), p300/CBP (CREB-binding protein), basal transcription factors, and nuclear receptor co-activators [2,3,5].

Among these, GNAT family have high similarity with yeast histone acetyltransferase Gcn5 that are typically representative of the GANT family have been extensively studied. GCN5 is functioned in the general regulation of amino acid synthesis signaling pathway and exhibits transcriptional-associated histone acetyltransferase [6,7,8]. Its activity seems to be dependent on the association in different multisubunit complexes, such as SAGA (Spt-Ada-Gcn5-Acetyltransferase), ADA (Ada2-Gcn5-Ada3), and SLIK/SALSA (SAGA-like) [9]. Gcn5 orthologues have been demonstrated to be required for development, differentiation, virulence, and regulation of autophagy in yeast and filamentous fungi [10,11,12,13,14,15,16,17]. The *Aspergillus flavus* Gcn5 protein is required for morphological development, aflatoxin biosynthesis, stress responses, and pathogenicity [14]. The Gcn5 of *Fusarium graminearum* is essential for sexual development, stress response, mycotoxin production, and virulence [17]. In the banana wilt disease causal agent *Fusarium oxysporum* f.sp. *cubense*, Gcn5 regulates growth, asexual sporulation (conidiation), and pathogenicity [16].

Another histone acetyltransferase Elp3 (Elongator protein 3), the catalytic subunit of the eukaryotic Elongator complex, is a lysine acetyltransferase that acetylates the C5 position of wobble-base uridines in tRNAs and this RNA acetylation of anticodon bases affects the ribosomal translation elongation rates [18]. Elp3 possesses a radical S-adenosyl methionine (rSAM)-binding domain and a lysine acetyltransferases (KAT) domain. The rSAM and KAT domains are held together by an extensive hydrogen bond network, forming a cleft between them to accommodate the tRNA substrate. Although there are variations in their primary amino acid sequences, the structure of the KAT domain of Elp3 resembles other acetyltransferases that all share a similar protein fold. The Elp3 KAT domain is most similar to Gcn5 proteins [19]. Elongator complex can physically interact with core histones and nucleosome [20]. However, the function of Elp3 in filamentous fungi has gained relatively little attention to date. Only one Elp3 has been reported in filamentous fungi, where *F*. *graminearum* Elp3 plays an important role in asexual and sexual development, oxidative stress response, trichothecene production, and virulence [21].

In the human pathogenic fungus *Aspergillus fumigatus*, two genes predicted to encode putative GNAT family HATs have been identified: *elp3* (Afu5g06140) and *gcn5* (Afu4g12650) (dbHimo: an Epigenetic Platform, hme.rice.snu.ac.kr, accessed on 17 November 2021). In *A*. *fumigatus*, Gcn5 (GcnE) is shown to regulate vegetative growth, conidiation, stress response, biofilm formation, and virulence of CEA17 strain [15,22]. In this study, we characterize functions of both GcnE and Elp3 in *A. fumigatus*-employing AF293 strain. The mutational inactivation of *elp3* and *gcnE* followed by a thorough analyses of the phenotypes indicate that the GNAT family HATs GcnE and Elp3 play a differential and pivotal role in governing conidiation, biofilm formation, stress response, toxin production, and virulence in *A*. *fumigatus*.

## 2. Results

### 2.1. Summary of Elp3 and GcnE in A. fumigatus

The predicted Elp3 protein consists of a 574 amino acid (aa) and has an Elp3 domain (109 to 371 aa) in the N-terminal region and an acetyltransferase 10 domain (511 to 566 aa) at the C-terminal region. The amino acid sequence of Elp3 of *A*. *fumigatus* shows 93.36~100% identity with Elp3-like proteins of other *Aspergillus* species. In the unrooted phylogenetic analysis based on the full-length amino acid sequences, Elp3 of *A*. *nidulans* is distinctly related (Figure 1A). On the other hand, GcnE has an N-terminal acetyltransferase 10 domain (94 to 207 aa) and a C-terminal bromodomain (292 to 400 aa). The bromodomain has been found in human, fruit fly, and budding yeast proteins, and is functionally linked to the HAT activity of co-activators in the transcriptional regulation of genes [23,24]. The amino acid sequence of GcnE (408 aa) shows 91.62~100% identity with GcnE found in phylogenetically related *Aspergillus* species (Figure 1B).

### 2.2. Elp3 and GcnE Are Required for Proper Vegetative Growth and Conidiaiton

The colony diameter of the Δ*elp3* and Δ*gcnE* mutants were considerably lower than that of WT in solid glucose minimal medium containing 0.1% yeast extract (MMY), and the Δ*gcnE* mutant showed abnormal colony color (Figure 2A). We also generated relevant complemented strains and found that they all showed the identical phenotypes with WT (Appendix A). The radial growth rate was also significantly decreased by the loss of *elp3* and *gcnE* (Figure 2B). In addition, the quantitative analyses of the number of conidia per growth area and per plate have revealed that the number of conidia in the Δ*elp3* mutant was slightly increased to about 134% and 144% compared to those of WT. In contrast, asexual spore production in the Δ*gcnE* mutant was significantly decreased to approximately 46% and 7.3% of those in the WT (Figure 2C). The levels of mRNA of the key asexual development regulators *abaA*, *brlA*, and *wetA* were also significantly decreased in the Δ*gcnE* mutant (Figure 2D).

### 2.3. Elp3 and GcnE Influence Biofilm Formation

To further investigate the functions of Elp3 and GcnE, we examined the effect of respective encoding gene loss on biofilm formation. As shown in Figure 3A, the Δ*elp3* and Δ*gcnE* mutants are markedly impaired in the formation of adherent biofilms on plastic plate at about 80% and 30% compared with WT strain, respectively. To identify the mechanism of hypo-adherence in mutant strains, we analyzed the mRNA levels of biofilm-formation associated genes, *medA*, *somA*, *stuA*, and *uge3*. The transcripts levels of most of these genes were significantly reduced in the mutants (Figure 3B). Scanning electron microscopy of the Δ*gcnE* mutant exhibited a complete loss of surface structure and intercellular matrix compared to WT and Δ*elp3* mutant (Figure 3C).

### 2.4. Elp3 and GcnE Affect Cell Wall Stress Response

We hypothesized that the differences in the hyphal architectures of the mutants affect cell wall stress response. To test this, we examined the effects of cell wall stressors on the growth of these strains by exposing them to cell-wall-damaging compounds (100 µg/mL), calcofluor white (CFW) and Congo red. The Δ*elp3* mutant showed relatively higher growth than WT and the Δ*gcnE* mutant when treated with CFW, and both mutants exhibited significantly increased resistance to Congo red (Figure 4A). These findings suggest that cell wall composition and/or integrity may be affected by mutational inactivation of *elp3* and *gcnE*. We further analyzed mRNA levels of the key chitin biosynthetic gene *gfaA*, of which mRNA level was increased by cell wall stress [25]. When induced with Congo red (100 µg/mL, for 6 h), the mRNA levels of *gfaA* were significantly increased in both mutants compared with WT (Figure 4B). These results indicate that Elp3 and GcnE are associated with proper cell wall biogenesis and cell wall integrity.

### 2.5. Elp3 and GcnE Functions in Oxidative Stress Response

To explore the roles of Elp3 and GcnE in response to oxidative stress, WT and mutant strains were incubated on YG supplemented with oxidative stress agents, 6 mM H_2_O_2_ and 50 µM menadione (MD). The Δ*elp3* mutant showed slightly reduced growth in response to both stress conditions. In contrast, the Δ*gcnE* mutant responded differentially to H_2_O_2_ and MD. Although the Δ*gcnE* mutant showed significantly high resistance against H_2_O_2_, it exhibited significant growth reduction in the presence of MD (Figure 5A). To further investigate the role Elp3 and GcnE, we analyzed activities of the ROS detoxifying enzymes. As shown in Figure 5B, while activities of conidia-specific catalase (CatA), mycelia-specific catalase (Cat1), and glutathione peroxidase (GPx) were observed in both mutants in the induction condition, the Cat1 activity was noticeably increased in the Δ*gcnE* strain. On the other hand, the activity of SOD1 in the Δ*gcnE* mutant was decreased in the induction condition compared to the control (Figure 5B). These results indicate that the elevated tolerance to H_2_O_2_ and reduced resistance to MD of the Δ*gcnE* mutant might be associated with high Cat1 and low SOD1 activities, respectively.

### 2.6. Elp3 and GcnE Affect Resistance to Triazol Antifungals

To determine whether Elp3 and GcnE are involved in triazole antifungal drug resistance, we performed E-strip tests with itraconazole and voriconazole. The loss of *gcnE* resulted in significantly increased sensitivity to itraconazole and voriconazole (Figure 6A). Triazoles bind to lanosterol 14-α demethylase (Cyp51/Erg11A) and inhibit the synthesis of ergosterol [26]. To investigate the mechanism of the sensitivity of the Δ*gcnE* mutant to triazoles, we determined the mRNA levels of the Cyp51-encoding genes *cyp51A* and *cyp51B*, as well as the *srbA* required for hypoxia response and azole drug resistance [27]. As shown in Figure 6B, while mRNA levels of these genes were significantly lower in both mutants than WT, the Δ*gcnE* mutant exhibited extremely low mRNA levels of all three genes, suggesting the critical role of GcnE in resistance to azole antifungals.

### 2.7. Elp3 and GcnE Oppositely Regulate Secondary Metabolites Production

As these GNAT HATs play important roles in various biological processes, we further examined the production of secondary metabolites in the culture extracts of WT, Δ*elp3*, and Δ*gcnE* strains. As shown in Figure 7A while the Δ*gcnE* mutant showed an increased production of several metabolites, including gliotoxin (GT), the Δ*elp3* mutant exhibited decreased GT production compared to WT. The production of numerous unidentified metabolites was also decreased in the absence of *elp3* compared to WT (Figure 7A). The high level of GT was associated with a significant increase in the mRNA level of *gliZ*, the C6 transcription factor required for the *gli* gene expression (Figure 7B).

### 2.8. GcnE Plays an Important Role in Virulence

In order to assess the pathological significance of Elp3 and GcnE, the conidia of three strains were intranasally introduced into immunocompromised mice and the pathological outcomes were analyzed by monitoring mouse survival. As shown in Figure 8A, mice infected with the Δ*gcnE* mutant spores survived significantly longer than those infected with the WT or Δ*elp3* spores (*p* value was 0.0176). Consistently, the loss of *gcnE* resulted in a significantly decreased pulmonary fungal burden of mice (Figure 8B). Next, we investigated the interaction of conidia to murine alveolar macrophage. The conidia of three strains were challenged with macrophages and the number of macrophages that ingested conidia after 4 h co-culture were counted. A significant increase in phagocytosis was observed in the Δ*gcnE* mutant conidia (14.26%) compared to those of WT and Δ*elp3* (Figure 8C). However, there were no differences in phagocytic index (average number of conidia per macrophage; data not shown). To further understand the basis for the differences in mouse survival, lung tissue sections were prepared from infected mice and stained with periodic acid–Schiff (PAS) to compare the extent of fungal impact and hyphal growth. As shown in Figure 8D, in sections of the lungs infected with the Δ*gcnE* mutant, PAS staining revealed a rather small number of fungal cells, which were similar to the negative control.

### 2.9. Targets of Elp3 and GcnE

To identify whether histone H3 acetylation levels were altered by the loss of *elp3* and/or *gcnE*, we performed Western blot analyses with specific antibodies against H3acK4, H3acK9, H3acK14, and H3acK27, with antibodies against H3 as a loading control. In the Δ*elp3* mutant the intensities of signals for all tested antibodies were increased compared with those of WT. However, in the Δ*gcnE* mutant, the signal intensities for H3acK9, H3acK14, and H3acK27 were remarkably decreased compared to WT (Figure 9). These results imply that GcnE might catalyze the acetylation of lysine at residues 9, 14, and 27 of histone H3. However, we could not find the target of Elp3.

## 3. Discussion

Histone acetylation has been established as a principal epigenetic regulatory mechanism for the formation of euchromatin, which affect multiple cellular processes in Eukarya [28,29,30,31]. Among type A HATs, we focused to characterize GNAT family HATs, Elp3, and GcnE in *A. fumigatus*. To elucidate the functions of these HATs, we constructed deletion mutants using AF293 as a background strain and analyzed their functions in vegetative growth, asexual sporulation, biofilm formation, stress responses, secondary metabolite production, and virulence.

The reduced acetylation in the Δ*elp3* mutant contributed to the sexual and asexual developmental and produced longer conidia compared with WT in *F*. *graminearum* [21,32]. GcnE is also required for conidiation but not essential for vegetative growth [10]. Although Elp3 and GcnE were not essential for growth, conidiation was oppositely regulated by Elp3 and GcnE in *A*. *fumigatus*. It has been shown that GcnE might be responsible for histone H3K9/K14 acetylation at the *brlA* promoter, necessary for *brlA* expression [10].

We found that both mutants down-regulated biofilm formation. Glucose epimerase, Uge3 plays an essential role in the formation of adherent biofilm and its expression is regulated by transcription factors, MedA, SomA, and StuA [33,34,35]. The expression levels of these genes were significantly reduced, and the hyphal structures were altered by the loss of *elp3* and/or *gcnE*. Apart from adherent biofilm, oxidative stress scavenging activity is an important virulent factor in *A*. *fumigatus*. GcnE has previously been reported to be important for oxidative stress response [15]. Surprisingly, the Δ*gcnE* mutant was remarkably more resistant to H_2_O_2_ than WT, and showed higher catalases and GPx activities. In contrast to our results, the previous study observed an increased susceptibility of the Δ*gcnE* mutant to H_2_O_2_ in the CEA17 strain background. This difference can be explained by the fact that the number, locations, and expression of transposable elements between AF293 and CEA17 are strikingly different, where the differences are correlated with H3K9me3 modifications and higher genomic variations among strains of AF293 background [36]. Moreover, Colabardini et al. [36] further showed that the AF293 strains from different laboratories differ in their genome contents and found a frequently lost region in chromosome VIII.

Whereas the mRNA levels of the *cyp51A* and *cyp51B* significantly reduced by the loss of *elp3* and *gcnE*, the Δ*gcnE* mutant showed greatly increased sensitivity to triazoles than that of the Δ*elp3* mutant, indicating that additional regulating mechanisms were involved in the sensitivity of the Δ*gcnE* mutant. It had been reported that Gcn5 of *Candida albicans* regulated the expression of ergosterol biosynthesis genes, including *ERG3*, *ERG250*, *ERG11*, *ERG13,* and the binuclear Zn2-Cys6 transcription factor *UPC2* involved in the sterol uptake [37,38]. From these, we supposed that the Δ*gcnE* strain may exhibit more sensitivity against triazoles not only by the reduced expression of ergosterol biosynthesis genes but also by decreased ergosterol uptake.

Previously, it was shown that histone acetylation plays only a minor role in the regulation of primary metabolism [39,40,41], but it plays an important role in governing secondary metabolic systems [42,43,44,45]. In many fungi, the production of secondary metabolites is associated with acetylation modifications. In *F*. *graminearum*, secondary metabolism (SM) associated key gene clusters were regulated in FgSAS3- and FgGCN5-dependent manners [17] and aflatoxin production was blocked by the decreased and/or absence of the acetyltransferase MYST3 and GcnE in *Aspergillus flavus* [14,46,47]. In *A*. *nidulans*, GcnE can increase the acetylation of H3K9 and H3K14 in the SM cluster and induce the synthesis of secondary metabolite orsellinic acid. The production of secondary metabolites is accompanied by a global increase in H3K14 acetylation and increased H3K9 acetylation was only found within gene clusters [43]. However, most of secondary metabolites were down-regulated in the Δ*elp3* mutant and up-regulated in the Δ*gcnE* mutant in our results, suggesting that the functions and acetylation targets of these enzymes might be different. GCN5 protein is the core part of the histone acetylation complex SAGA (SPT-Ada-GCN5 acetyltransferase) and functions at specific lysine residues in histones H3 and H2B, regulating global gene expression [48,49]. ELP3 is the catalytic subunit of the multisubunit Elongator complex associated with elongating RNA polymerase II (RNAPII) [50]. It consists of six subunits (ELP1 to ELP6). In yeast, purified ELP3 acetylated histones in vitro and the HAT activity of ELP3 was essential for its function in vivo [51]. Human ELP3 was also acetylate histones H3 and H4 [52]. Although the biochemical function of Elp3 is conserved in eukaryotes, the targets can vary by species.

Although the Δ*gcnE* mutant was more resistant to H_2_O_2_ and produced large amounts of secondary metabolite, including GT, the virulence in a neutropenic murine model was significantly lower than that of WT and Δ*elp3* strain. Similarly, the *gcn5* mutant of human pathogenic yeast *Cryptococcus neoformans* was avirulent in an animal model [13] and Gcn5 of *Candida albicans* was required for the pathogenesis in a murine model [53]. The Δ*gcnE* mutant’s attenuated virulence may be due to the down-regulation of biofilm formation, thus up-regulating the phagocytosis. Our histological analysis has revealed very low spore germination and the invasion of hyphae in the lungs of mice infected with the Δ*gcnE* mutant, suggesting that GcnE might be a key factor determining the fungal virulence.

Numerous acetylation sites were established in histone H3, highly conserved lysine residues, and these are important for biological process [54,55,56]. The synthesis of the secondary metabolites was accompanied by the increased acetylation of H3K14, whereas the increased acetylation of H3K9 occurred only within secondary metabolism gene clusters and the acetylation of H3K14 was more important than the acetylation of H3K9 in *A*. *nidulans* [43,44,53]. H3K4 was methylated and acetylated in budding yeast, depending on Gcn5 and Rtt109 [57]. Gcn5 is responsible for the acetylation of H3K9 and H3K14 in yeast and many other filamentous fungi. The GcnE of *A*. *flavus* was required for inducing aflatoxin biosynthesis via the acetylation of H3K14 in the promoter regions of aflatoxin genes [14]. It has been reported that *FgGCN5* was essential for the acetylation of H3, H3K9, H3K14, H3K18, and H3K27 [17]. In our results, GcnE also exhibited a significant impact on the acetylation of H3K9, H3K14, and H3K27 but not H3K4. The Elongator complex of budding yeast predominantly acetylated H3K14, and the acetylation level was markedly reduced by the loss of *elp3* in *F*. *graminearum* [20,21]. However, we could not find the target of Elp3 of *A*. *fumigatus* and the identification of the target requires further investigation.

In conclusion, we have revealed that GNAT family HATs Elp3 and GcnE are associated with diverse biological processes, including vegetative growth, asexual sporulation, stress response, secondary metabolites production, and virulence in the human pathogenic fungus *A*. *fumigatus*. Importantly, we anticipate that our findings of the necessity of both for biofilm formation and of GcnE for triazole antifungal resistance and may elucidate a new road to a novel antifungal drug discovery.

## 4. Materials and Methods

### 4.1. Bioinformatic Analysis

The amino acid sequences of *A*. *fumigatus* Elp3 and Gcn5 were retrieved from the *A*. *fumigatus* AF293 genomic database (http://www.aspergillusgenome.org/, accessed on 4 December 2021). Amino acid sequences of full length of Elp3 and GcnE were subject to the SMART program (http://smart.embl-heidelberg.de, accessed on 25 January 2022) for structural comparison. The phylogenetic trees were constructed using the neighbor-joining method. The percentage of replicate trees in which the associated taxa clustered together in the bootstrap test (500 replicates) are shown next to the branches and evolutionary analyses were conducted in MEGA X software (http://www.megasoftware.net, accessed on 25 January 2022).

### 4.2. Strains and Culture Conditions

All *A. fumigatus* strains used in this study (Table 1) were derivatives of the WT AF293 strain [58]. Fungal strains were grown on glucose minimal medium (MMG) or MMG with 0.1% yeast extract (MMY) and appropriate supplements, as described previously [59].

### 4.3. Construction of Mutant Strains

The deletion construct generated by double-joint PCR [61] containing the *A*. *nidulans* selective marker *AnipyrG* with the 5′ and 3′ flanking regions of the *A*. *fumigatus elp3* gene (Afu5g06140), or the *gcnE* gene (Afu4g12650) was introduced into the recipient strains [62]. The selective marker was amplified from *A. nidulans* FGSC A4 genomic DNA. To complement the mutants, a double-joint PCR method was used [61] with *hygB* as a selective marker. The null mutants and complemented strains were confirmed by diagnostic PCR (using primer pairs oligo1472/1473 and oligo1478/1479) followed by restriction enzyme digestion (Appendix A). The oligonucleotides used in this study are listed in Appendix A.

### 4.4. Nucleic Acid Manipulation and RT-qPCR Analysis

Total RNA isolation and RT-qPCR were performed, as previously described [63,64,65]. RT-qPCR was performed using One-Step RT-PCR SYBR Mix (Doctor Protein, Seoul, Republic of Korea) and a Rotor-Gene Q real-time PCR system (Qiagen, Hilden, Germany). The amplification of a specific target DNA was verified by melting curve analysis. The expression ratios were normalized to the expression level of the endogenous reference gene *ef1α* [66,67] and calculated by the 2^−ΔΔCq^ method [68]. The expression stability of *ef1α* and efficiencies of PCRs of the target genes were determined, as previously described [69]. Expression levels of target gene mRNAs were analyzed using appropriate oligonucleotide pairs (Appendix A).

### 4.5. Phenotype Experiments

Radial growth was assayed by inoculation of spores in the center of appropriate media and measurement of colony diameters every 24 h, and data are presented as the means of triplicates. Conidial production was quantified from two inoculation methods. Point-inoculated cultures were used as per the growth area and overlay-inoculated cultures were used on each plate. Conidia were collected using 0.5% Tween 80 solution, filtered through Miracloth (Calbiochem, San Diego, CA, USA), and counted using a hemocytometer. Biofilm formation assays were performed as previously described with some modifications [34]. Briefly, 2 mL of RPMI 1640 broth containing 2 × 10^5^ conidia were inoculated into 12-well untreated plates at 37 °C for 24 h. Then, the culture medium was removed, and the wells were washed three times with phosphate-buffered saline (PBS). Biofilms were visualized by applying 2 mL of 0.05% crystal violet solution for 5 min. Excess stain was removed, and the plates were washed 3 times with 3 mL of PBS. The stain remaining in the biofilm was extracted by adding 1 mL of ethanol. Biofilm density was determined by measuring the absorbance of the destained solution at 570 nm. To test for oxidative and cell wall stress, menadione (MD, 50 µM), H_2_O_2_ (6 mM), calcofluor white (CFW, 100 µg/mL), and Congo red (100 µg/mL) were added to the YG medium after autoclaving. The effects of antifungal agents on the growth of the WT and mutant strains were investigated using E-test strips (Biomérieux, Durham, NC, USA). Conidial suspensions (1 × 10^4^ conidia) were inoculated into a solid RPMI 1640 medium, and E-test strips were placed on the plate. After incubation at 37 °C for 48 h, MIC values were determined as the zone edge intersecting the strips. The production of secondary metabolites and GT was determined, as described previously [69,70]. Briefly, 1 × 10^7^ conidia/mL of each strain were inoculated into 5 mL complete medium in a test tube and incubated at 37 °C slanted for 7 days. GT was extracted with chloroform. Individual chloroform extracts were air-dried and resuspended in 100 µL of methanol. Aliquots (10 µL) of each sample were applied to a TLC silica plate containing a fluorescence indicator (Kiesel gel 60, E. Merck, Burlington, MA, USA). The TLC plate was developed with chloroform:methanol (9:1, *v/v*). GT standard was purchased from Sigma-Aldrich (Burlington, MA, USA).

### 4.6. Enzyme Assay and Western Blot Analysis

To determine the activities of catalase, glutathione peroxidase (GPx), and SOD, the conidia (1 × 10^5^) of relevant strains were inoculated into liquid YG and incubated at 37 °C, 250 rpm for 18 h. Then, oxidative stress agents were added and the conidia were incubated up to 24 h. The mycelia were disrupted with glass beads in 20 mM phosphate buffer (pH 7.5) supplemented with a protease inhibitor cocktail. Protein content was quantified using Bradford reagent (Bio-Rad Laboratories, Inc., Hercules, CA, USA) and bovine serum albumin as a standard. Catalase and GPx activities were visualized by negative staining with ferricyanide [71,72]. SOD activity was determined as the inhibition of nitroblue tetrazolium reduction [73]. For Western blotting, histone was extracted with Histone Extraction Kit (Abcam, Cambridge, UK, 113476), according to manufacturer’s manual. Approximately 50 µg of nuclear protein extract was electrophoresed on a 10% SDS-PAGE gel and subsequently electroblotted to nitrocellulose membranes. Relevant histone modifications were detected with primary antibodies specific to histone H3 (Abcam, 1791), H3acK4 (Abcam, 176799), H3acK9 (Abcam, 177177), H3acK14 (Abcam, 203952), and H3acK27 (Abcam, 4729) antibodies. Relative intensities of the enzyme activities and Western blot were quantified using the Image J 1.52k software (NIH, Bethesda, MD, USA).

### 4.7. SEM

For SEM analysis, hyphae were grown for 24 h in phenol red-free RPMI 1640 medium on glass coverslips, fixed with 2% glutaraldehyde and 2% paraformaldehyde in 50 mM cacodylate buffer (pH 7.4) for 1 h at 4 °C. The samples were sequentially dehydrated in ethanol each for 10 min and then kept in a mixture of 100% ethanol and isoamyl acetate (2:1, 1:1, 1:2) for 10 min and finally in pure isoamyl acetate for 15 min. After removal of isoamyl acetate, the samples were treated with HMDS (hexamethyldisilazane) for 40 min. Samples from which HMDS was removed were naturally dried for about 1 h and then were sputter-coated with a thin layer of gold. Samples were viewed under SUPRA 55VP scanning electron microscope (Carl Zeiss, Germany) at an accelerating voltage of 3 kV in the Korea Basic Science Institute (Chuncheon, Republic of Korea).

### 4.8. Murine Virulence and Phagocytosis Assay

The virulence assay was conducted, as previously described [69,74,75]. Briefly, the CrlOri: CD1 (ICR) (Orient Bio Inc., Seongnam, Republic of Korea) female mice (6–8 weeks old, weighing 30 g) were immunosuppressed by the treatment with cyclophosphamide and cortisone. Mice were anesthetized with isoflurane and then intranasally infected with 1 × 10^7^ conidia of *A*. *fumigatus* strains (10 mice per each fungal strain) suspended in 30 µL of 0.01% Tween 80 in PBS. Mice were monitored every 12 h for 8 days after the challenge. Control mice used in all experiments were inoculated with sterile 0.01% Tween 80 in PBS. For histology experiments, the mice were sacrificed at 3 day after conidia infection. Kaplan–Meier survival curves were analyzed using the log-rank (Mantel–Cox) test for significance. A phagocytic assay was performed according to a modified method [76,77]. The MH-S cell lines were maintained in RPMI 1640 containing 10% fetal bovine serum (Invitrogen, Carlsbad, CA, USA) and 50 μM of 2-mercaptoethanol (Sigma, St. Louis, MO, USA). The MH-S cells were adhered to coverslips in 6-well plates at a concentration of 5 × 10^5^ cells/mL for 2 h and subsequently challenged with 1.5 × 10^6^ conidia for 1 h. Unbound conidia were removed by washing with PBS and then incubated for 2 h at 37 °C in an atmosphere of 5% CO_2_. Wells were then washed with PBS and observed with microscopy. The percentage of phagocytosis and the phagocytosis index were assessed.

### 4.9. Statistical Analysis

All experiments performed in triplicate and *p* < 0.05 was considered a significant difference. Data were expressed as mean ± standard error. GraphPad Prism 4 (GraphPad Software, Inc., San Diego, CA, USA) was used for the statistical analyses and graphical presentation of survival curve.

## Figures and Tables

**Figure 1 ijms-24-02179-f001:**
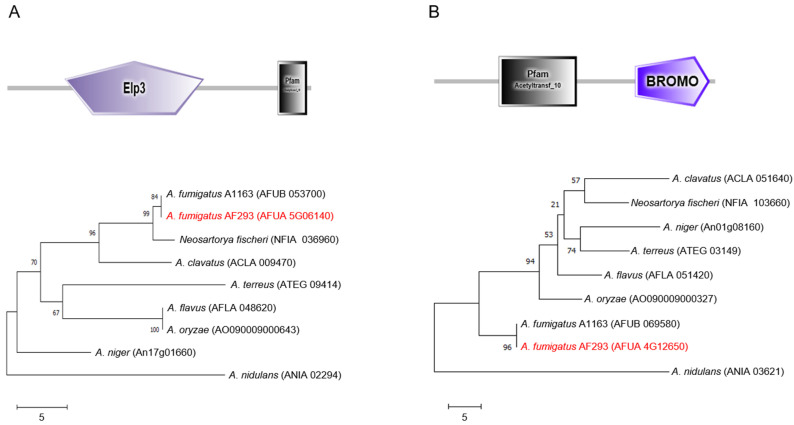
Domain architecture and phylogeny of the Elp3 and GcnE proteins. (**A**) A schematic presentation of the domain structure of Elp3 and a phylogenetic tree of Elp3-like proteins in various *Aspergillus* species constructed based on the matrix of neighbor-joining distances. Numbers listed next to the branches indicate bootstrap values. (**B**) A domain structure of the GcnE and a phylogenetic tree of GcnE orthologs in various *Aspergillus* species. Domain structures are presented using SMART (http://smart.embl-heidelberg.de, accessed on 4 May 2022). The target genes of this study are presented in red.

**Figure 2 ijms-24-02179-f002:**
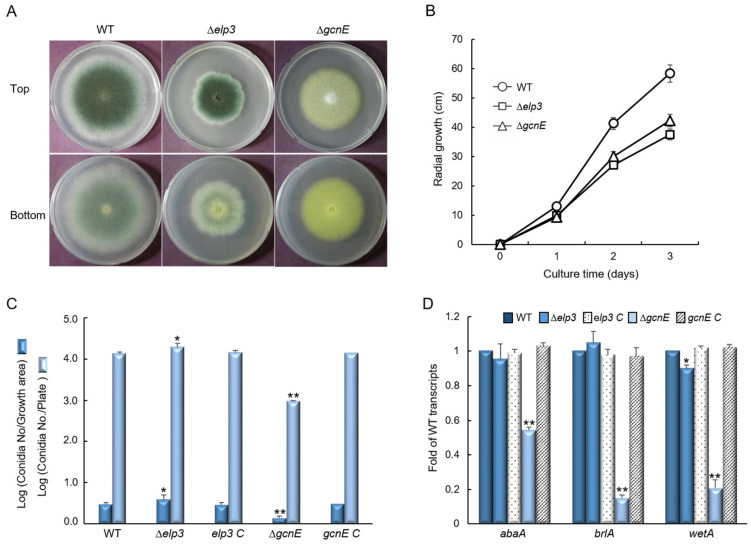
Differential roles of Elp3 and GcnE in vegetative growth and conidiation. (**A**) Colony photographs of WT, Δ*elp3*, and Δ*gcnE* strains point-inoculated and grown in solid MMY. (**B**) Radial growth of five strains grown on solid MMY for 3 days determined by colony diameter. (**C**) Conidia numbers produced by each strain per growth area and per plate. (**D**) Transcript levels of the key asexual developmental regulators in the mutants and complemented strains relative to those in WT at 3 days determined by quantitative RT-PCR (RT-qPCR). Fungal cultures were grown in MMY, and mRNA levels were normalized to the expression level of the *ef1α* gene. Statistical significance of differences was assessed by Student’s *t*-test: * *p* < 0.05, ** *p* < 0.01.

**Figure 3 ijms-24-02179-f003:**
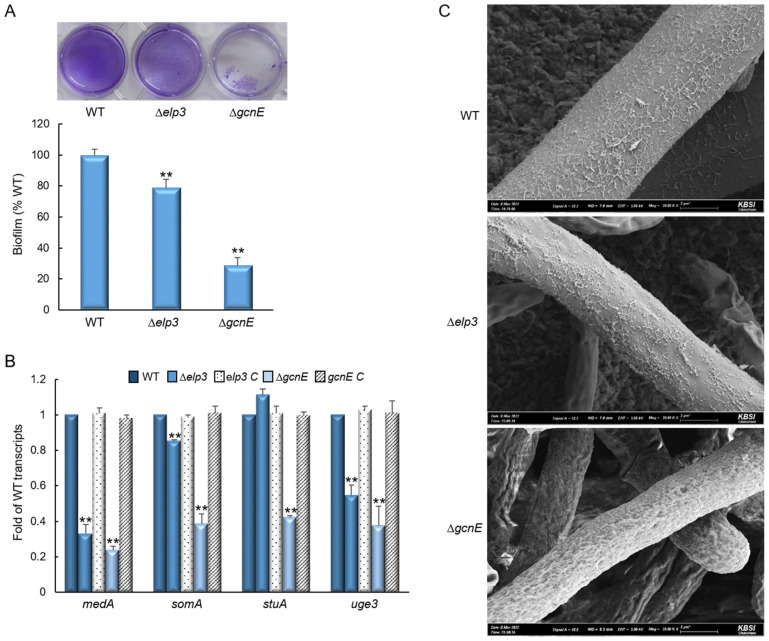
Elp3 and GcnE are required for biofilm formation and adherence to plastic. (**A**) Biofilm formation by hyphae of the indicated strains after 24 h growth on polystyrene plates. After washing, hyphae were stained with crystal violet for visualization and relative biofilm formation levels to WT were calculated. (**B**) Transcript levels of the key biofilm formation genes in the mutants and complemented strains relative to those in WT determined by RT-qPCR. The mRNA levels were normalized to the expression level of the *ef1α* gene. (**C**) Scanning electron micrographs of hyphae of three strains after 24 h of growth. Magnification was 20,000×. Statistical significance of differences was assessed by Student’s *t*-test: ** *p* < 0.01.

**Figure 4 ijms-24-02179-f004:**
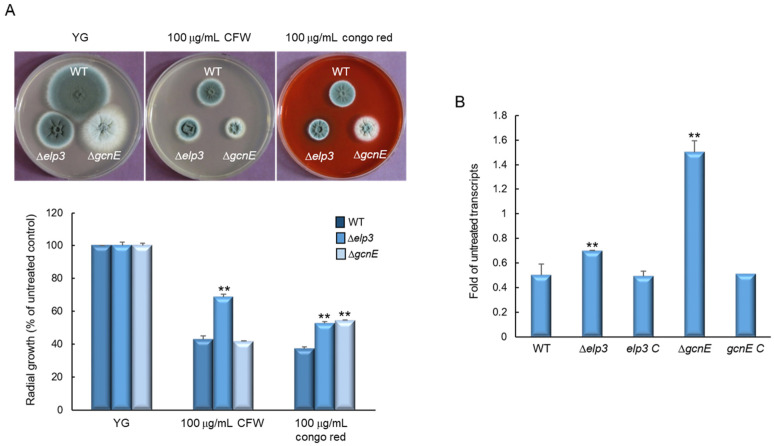
Elp3 and GcnE affect sensitivity to cell-wall-perturbing agents. (**A**) Colony appearance and radial growth inhibition after inoculation of 1 × 10^5^ conidia on YG (yeast extract glucose) containing cell wall stressors. (**B**) Transcript levels of the key chitin biosynthetic gene *gfaA* in the mutants and complemented strains relative to the corresponding level in the WT strain determined by quantitative RT-PCR (RT-qPCR). The mRNA levels were normalized to the expression level of the *ef1α* gene. Statistical significance of differences was assessed by Student’s *t*-test: ** *p* < 0.01.

**Figure 5 ijms-24-02179-f005:**
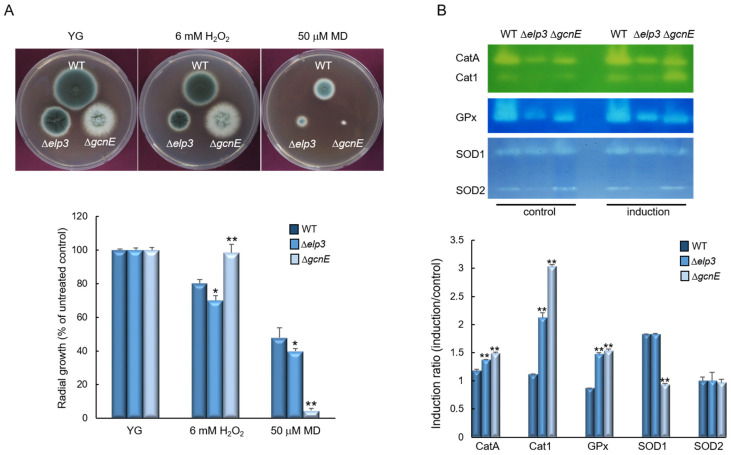
Differential roles of Elp3 and GcnE in response to oxidative stress. (**A**) Colony appearance and radial growth inhibition after inoculation of 1 × 10^5^ conidia on solid YG-containing oxidative stressors. (**B**) Catalase, glutathione peroxidase (GPx), and SOD activity of the WT and mutant strains. Induction ratios of each enzyme’s activity are shown below. Statistical significance of differences between WT and mutant strains was evaluated using Student’s *t*-test: * *p* < 0.05, ** *p* < 0.01.

**Figure 6 ijms-24-02179-f006:**
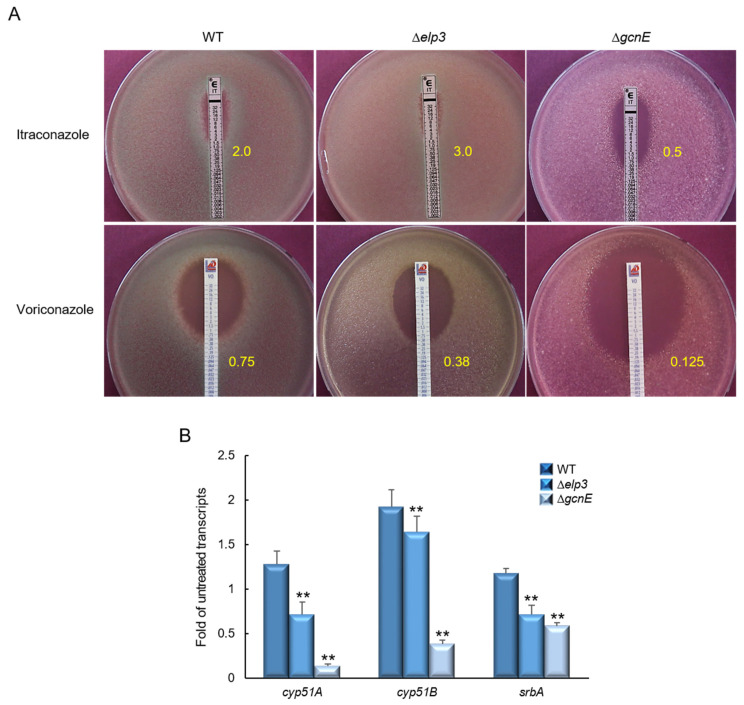
Differential roles of Elp3 and Gcn5 in resistance to triazole antifungal agents. (**A**) Determination of the effects of antifungal agents on the growth of WT and mutant strains by using E-test strips. Minimal inhibitory concentrations are indicated in yellow numbers. (**B**) RT-qPCR analysis of *cyp51A*, *cyp51B*, and *srbA* expression levels in mutant and complemented strains compared to that of WT. Statistical significance of differences in expression levels was evaluated using Student’s *t*-test: ** *p* < 0.01.

**Figure 7 ijms-24-02179-f007:**
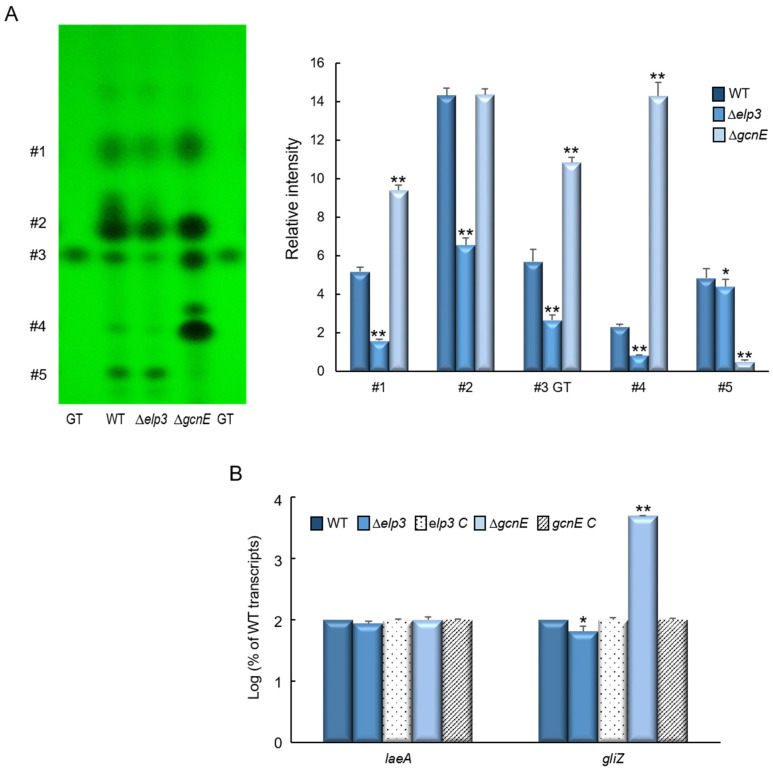
Roles of Epl3 and GcnE in the production of secondary metabolites. (**A**) Determination of GT and several secondary metabolites in WT and mutant strains. Left: a representative thin layer chromatography (TLC) of the culture supernatant of each strain extracted with chloroform. Right: a graph of relative intensities of individual chromatogram spots. (**B**) RT-qPCR analysis of mRNA levels of *laeA* and *gliZ* in mutant and complemented strains compared to those in WT. Statistical significance of differences between WT and mutant strains was assessed by Student’s *t*-test: * *p* < 0.05, ** *p* < 0.01.

**Figure 8 ijms-24-02179-f008:**
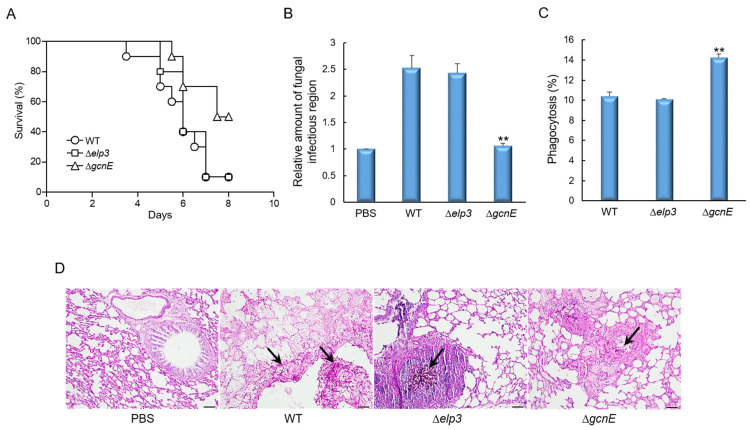
Effects of Epl3 and GcnE on the virulence of *A*. *fumigatus*. (**A**) Survival curves of mice intranasally administered with conidia of WT and mutant strains (n = 10/group). (**B**) Fungal burden in the lungs of mice infected with T or mutant strains. (**C**) Phagocytosis of WT and mutant strains. Phagocytosis indicates percentage of macrophages containing one or more ingested conidia (n = 20). (**D**) Representative lung sections of mice from different experimental groups stained with periodic acid–Schiff reagent (PAS). Arrows indicate fungal mycelium. Statistical significance of differences between WT and mutant strains was evaluated by Student’s *t*-test: ** *p* < 0.01.

**Figure 9 ijms-24-02179-f009:**
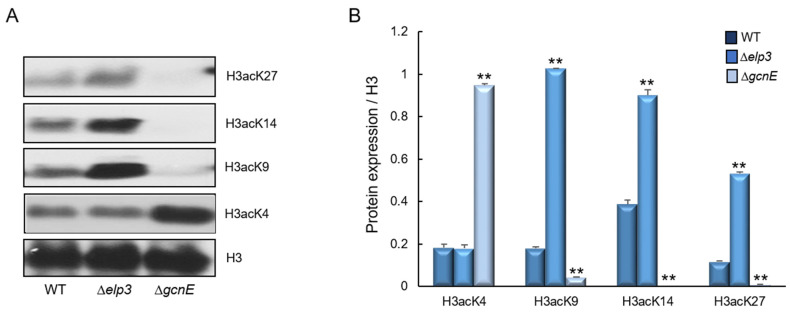
Western blot analysis of histone H3 acetylation levels. (**A**) The anti-acetyl H3K4 (H3acK4), anti-acetyl H3K9 (H3acK9), anti-acetyl H3K14 (H3acK14), and anti-acetyl H3K27 (H3acK27) antibodies were used for the detection of alterations of acetylation levels. Antibody against H3 was used as a loading reference. (**B**) Quantification of Western blot signals in triplicates. Data were expressed as mean (relative to H3) ± standard error. Statistical significance of differences between WT and mutant strains was evaluated by Student’s *t*-test: ** *p* < 0.01.

**Table 1 ijms-24-02179-t001:** *A*. *fumigatus* strains used in this study.

Strain	Genotype	Reference
AF293	Wild type	[58]
AF293.1	*pyrG1*	[60]
Δ*Afuelp3*	Δ*Afuelp3*::*AnipyrG*; *AfupyrG1*	This study
Δ*AfugcnE*	Δ*AfugcnE*::*AnipyrG*; *AfupyrG1*	This study
*Afuelp3* *C*	Δ*Afuelp3*::*AnipyrG*;*AfupyrG1*;*Afuelp3*::*hygB*	This study
*AfugcnE C*	Δ*AfugcnE*::*AnipyrG*;*AfupyrG1*;*AfugcnE*::*hygB*	This study

## Data Availability

The data presented in this study are available on request from the corresponding authors.

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
