# Peer review of "Functional Characterization of the GNAT Family Histone Acetyltransferase Elp3 and GcnE in Aspergillus fumigatus"

_ijms, 2023, doi:10.3390/ijms24032179_

Round 1
Reviewer 2 Report
Choi et al. work on “Functional Characterization of the GNAT Family Histone 2 Acetyltransferase Elp3 and GcnE in Aspergillus fumigatus” is interesting. The study is appreciable, the context is well presented, the approach fairly adequate and the results deserve publication. However, there are certain points which need to be addressed. The authors must address all the issues for it to be considered for the publication.
1. The abstract should reflect the problem, the method, and the conclusions. It can be improved further.
2. Introduction is not sufficient. Please add more related papers; clearly state the importance of the topic, recent examples from the literature. And in the last paragraph of the Introduction, please state your motivation, goals and objectives, the novelty of this research, and potential contribution to the literature.
3. In section 2.1, the method employed for generating phylogenetic tree in MEGA7 software must be specified and explained clearly.
4. In page 4 section 2.3 on line 116, the authors state that the surface decoration has decreased in the Δelp3 mutant, based on the SEM image in figure 3c. However, the significant difference in surface structure couldn’t be observed between WT and Δelp3. Explain in detail.
5. In page 11 line 260, the author says that “Surprisingly, the ΔgcnE mutant was remarkably more resistant to H2O2 than WT….. in CEA17 strain background”. The authors should explain what could be the reason behind this contrast observation? Authors should explain about this?
6. There are several typo errors and missing prepositions and articles. Check throughout the manuscript and correct accordingly.
Ex: on line 145, write “oxidative stress” instead of “oxidative of stress”
On line 173, write “than WT” instead of “that WT”
On line 243, write “GNAT family” instead of “GANT family”
On line 258, write “loss of elp3” instead of “loss elp3”
7. On line 273 & 274, the authors suggest that the functions and acetylation targets of Elp3 and GcnE might be different. Explain more about the regulation of these enzymes with support from the literature.
8. Conclusion can be improved further. Include major outcome of your study in a succinct manner.

Reviewer 3 Report
This study focused on identification of GNAT family histone acetyltransferase Elp3 and GcnE in pathogenic fungus Aspergillus fumigatus which could cause infections in immunocompromised patients. Histone acetyltransferases play a critical role in diverse biological processes, like regulation of gene expression, and could be the targets of antifugal treatment. This study generated and characterized Δelp3 and ΔgcnE knockout mutants in A. fumigatus, and revealed the function of Elp3 and GcnE in morphological development, differentiation, virulence, and stress responses.
Recently, more and more evidences revealed that Elp3 primarily acts as a tRNA acetyltransferase in the elongator complex which is required for tRNA modifications in vivo. And in this study, the acetylation level increased in Δelp3 knockout mutant in A. fumigatus. The authors should carefully discuss histone acetyltransferase activity of Elp3 and make a clear conclusion.
1. In line 53, ‘acetyl transferase’ should be ‘acetyltransferase’.
2. In panel B of figure 2, there is no error bar. Did authors only measure one sample for each strain? It would be helpful to provide more measure data for all three strains and make the error bar.
3. In line 185, what’s the meaning of TLC? It would be helpful to provide the full name at the first time and describe this method in method section.
4. How to detect the metabolite gliotoxin? The authors should be provide more details of the detection method and describe the results clearly. And in panel A of figure 7, band of gliotoxin looks bigger than the bands of culture extracts of WT, Δelp3, and ΔgcnE strain. What’ s the reason? Will it still support the conclusion?
5. In panel A of figure 7, what’s the #4 band? It obviously increased in ΔgcnE strain.
6. In the Δelp3 mutant of F. graminearum, the acetylation reduced. It’s different to phonotype of Δelp3 mutant in A. fumigatus. Will this diversity of Elp3 in different species still support its activity of histone acetyltransferase?
Round 2
Reviewer 1 Report
The authors have satisfied all the queries I had in the previous version. I do not have any more comments/suggestions.
Reviewer 3 Report
The authors have answered all my questions clearly. I have no more comment.